# Registered report: Stress testing predictive models of ideological prejudice

Jordan L. Thompson[1]*, Abigail L. Cassario[2], Shree Vallabha[3], Samantha A. Gnall[1], Sada Rice[1], Prachi Solanki[2], Alejandro Carrillo[2], Mark J. Brandt[2], Geoffrey A. Wetherell[1]

**1** Department of Psychology, Florida Atlantic University, Boca Raton, Florida, United States of America,
**2** Department of Psychology, Michigan State University, East Lansing, Michigan, United States of America,
**3** FLAME University, Pune, India

* jordanthomps2021@fau.edu

## Abstract

In this registered report, we stress-tested existing models for predicting the ideology-prejudice association, which varies in size and direction across target groups. Previous models of this relationship use perceived ideology, status, and choice in group membership of target groups to predict the ideology-prejudice association across target groups. These analyses show that models using only the perceived ideology of the target group are more accurate and parsimonious in predicting the ideology-prejudice association than models using perceived status, choice, and all three characteristics in one model. Here, we stress-tested the original models by testing the models' predictive utility with new measures of explicit prejudice, a comparative operationalization of prejudice, the Implicit Association Test (IAT), and additional target groups. In Study 1, we directly tested the previous models using absolute measures of prejudice that closely resemble the measures used in the original study. Our results indicated that the models replicate with distinct, yet conceptually similar measures of prejudice. As in previous work, our ideology-only and ideology, status, and choice models were the best predictors of the ideology-prejudice association. In Study 2, we developed new ideology-prejudice models for a comparative operationalization of prejudice using both explicit measures and the Implicit Association Test. We tested these new models using data from the Ideology 2.0 project collected by Project Implicit. Our results indicate that this model-building strategy was not effective for relative or IAT prejudice measures. We found no significant differences in predictive ability between the models. These results indicate that the ideology-only and ideology, status, and choice models are effective in predicting the ideology-prejudice association in a variety of absolute prejudice measures, but our results suggest this may not generalize to relative or IAT measures.

**Data availability statement:** All data files are available on the project's OSF page: https://doi.org/10.17605/OSF.IO/BUWP7.

**Funding:** The author(s) received no specific funding for this work.

**Competing interests:** The authors have declared that no competing interests exist.

## Introduction

How can we predict whether liberals or conservatives will express more prejudice toward a group? Brandt [1] sought to predict the ideology-prejudice association for multiple target groups with models using perceived ideology, perceived group status, and perceived choice of group membership as predictors. The goal was to create models that make accurate predictions of the ideology-prejudice association toward a variety of target groups and to help us understand which theoretical perspectives are most useful when predicting ideological prejudice. We replicated and extended this work here.

In effort to forecast prejudice toward various groups, Brandt [1] focused on three main predictors of the ideology-prejudice association: the ideological position of a group, the group's status, and the extent to which group membership is a choice. First, research indicates people dislike groups that hold ideological positions dissimilar from their own [2–4], suggesting an association between perceived ideological dissimilarity and prejudice. For example, Chambers and colleagues found that conservatives prefer traditionally conservative groups, and liberals prefer liberal groups [3]. Second, other work suggests that conservatives may be more prejudiced in general [5], or conservatives may be more prejudiced against low-status groups [6]. For example, research suggests that people may be prejudiced toward privileged groups as well as marginalized groups [6], and political conservatism relates to prejudice against a variety of low-status targets [7]. Third, another perspective proposes that perceived choice regarding group membership is valued by conservatives as it helps define boundaries between groups [7,8]. Additional work indicates that people with lower levels of cognitive ability tend to be more prejudiced toward groups with little choice in group membership [9]. Brandt [1] compared these three perspectives to one another. In this project, we built upon and stress-tested Brandt's work [1] to try and predict the size and direction of the ideology-prejudice association.

The prejudice and ideology literatures do not offer many concrete methodological suggestions for predicting the magnitude and direction of the effect of ideology on prejudice against specific groups. As an example, the widely cited dual process models of prejudice suggest that a desire for social conformity and belief in a dangerous world predict right-wing authoritarian (RWA) attitudes, whereas a belief that the world is a competitive jungle predicts social dominance, both of which then predict prejudice [10]. Despite providing a theoretical rationale for the relationship between manifestations of ideology (i.e., RWA) and prejudice towards certain groups, the dual process model does not provide the scaffolding to predict the specific level of prejudice a person will exhibit towards a variety of target groups. Moreover, this work examines a limited set of low-status target groups. This is a gap that should be filled because there is both practical and theoretical value in models that can predict the ideology-prejudice association across targets [1].

First, a predictive model can help scholars examine the nature of the association between ideology and prejudice (i.e., the strength and direction of the relationship) depending on different characteristics of a target group (e.g., status, perceived

ideology). Different combinations of characteristics may yield different predictions of the strength and direction of the ideology-prejudice association, which allows for greater predictive accuracy.

Second, these models can be used to make predictions using new samples and target groups. This underscores the generalizability of such models, allowing scholars to anticipate associations before data are collected. Ideology-prejudice association predictions provide estimates of effect size, which means they can be used to conduct power analyses prior to collecting data and serve as a theoretical starting point when proposing new models. Likewise, predictive models have theoretical implications because predictions of effect size can be useful in falsifying hypotheses and comparing competing theoretical models. Indeed, recent research in psychology has used predictive models to test diverse outcomes, such as changes in trust [11], romantic interest [12], and success in psychotherapy [13]. Here we further developed predictive models of the ideology-prejudice association.

### Predicting ideological prejudice

Brandt [1] built models of the ideology-prejudice association using data from the 2012 wave of the American National Election Studies (ANES) and then tested the predictive accuracy of those models in new samples with new target groups. The ideology-prejudice association was estimated for 24 groups in the 2012 ANES, including political groups, religious groups, socioeconomic groups, and racial/ethnic groups. Each target group was also rated by a separate sample in terms of their perceived ideology, status, and choice in group membership. These group characteristics were then used to build models of the ideology-prejudice association. For example, an ideology-only model used only the perceived ideology of the target group to predict the size and direction of the ideology-prejudice association. Other models included other group characteristics (e.g., a status-only model) or combinations of characteristics (e.g., ideology, status, and choice). The estimates from these models were used to predict the size and direction of the ideology-prejudice association in various additional samples and target groups.

Brandt compared the predicted association with the observed association between ideology and prejudice using Mean Squared Errors (MSE) [1]. Specifically, to generate predicted values, he used data from the 2012 ANES to estimate the ideology-prejudice association for 24 groups. Then, he built models using measures of each group's perceived ideology, status, and choice (obtained from separate samples). The estimates from these models were used as the *predicted* ideology-prejudice association. The *observed* ideology-prejudice associations were obtained by estimating the ideology-prejudice association for each target group in each of four studies. Then, Brandt compared the predicted ideology-prejudice associations with the observed ideology-prejudice associations (residuals), squared them, and found their average for each model. The model with the lowest MSE was the best-fitting model (see Table 1 for the four models we focus on, plus a null model).

Table 1 includes Brandt's original equations [1]. The MSEs and SDs for each model and for each type of measure taken from our data are also presented in the table. Lower MSEs are more accurate.

Brandt's [1] predictive analyses revealed that models that included only ideology or included ideology, status, and choice group characteristics were the most accurate in predicting the ideology-prejudice association in new data. These models were more accurate than models that only included status or only included choice (for a replication, see [14]). Brandt [1] also included conventionalism, but because perceived conventionalism and ideology were highly overlapping, he focused on the ideology models. We did the same here. This pattern of results suggests the ideology-only model, the most accurate and most parsimonious model, was the best model because it yielded the lowest MSE values. Although perceptions of group status and choice have often been related to prejudice [5,8,15–17], the perceived ideology of the target group may be the biggest factor in understanding which groups liberals and conservatives express the most prejudice toward.

### Stress-testing existing models

The purpose of this project was to stress-test Brandt's [1] original models using alternative measures of prejudice, which include alternative explicit measures (i.e., relative measures of explicit prejudice) and reaction time measures (i.e., Implicit

Table 1. Predictive Equations of Prejudice from Brandt's [1] Work and MSEs (Study 1).

| Model Name | Theoretical Implication | Model | MSE (SD) Estimate Across Outcomes | MSE (SD) for Actual Prejudice Outcome | MSE (SD) for Gut Prejudice Outcome | MSE (SD) for Positive Prejudice Outcome | MSE (SD) for Negative Prejudice Outcome | Correlation Between Predicted and Observed Values (r) |
|---|---|---|---|---|---|---|---|---|
| ideology-only | Ideological differences explain ideology-prejudice association | $\hat{y}=0.022–1.420(\text{ideology})$ | .01 (.01) | .01 (.01) | .02 (.03) | .02 (.03) | .01 (.02) | .97 |
| status-only | Status differences explain ideology-prejudice association | $\hat{y}=0.001–0.846(\text{status})$ | .08 (.12) | .12 (.19) | .14 (.23) | .05 (.07) | .04 (.06) | .45 |
| choice-only | Choice differences explain ideology-prejudice association | $\hat{y}=0.041–0.398(\text{choice})$ | .09 (.12) | .13 (.20) | .16 (.24) | .05 (.07) | .05 (.06) | .17 |
| ideology, status, and choice | A combination of group characteristics explains ideology-prejudice association | $\hat{y}=0.016–1.505(\text{ideology}) + 0.128(\text{status}) + 0.072(\text{choice})$ | .01 (.01) | .02 (.02) | .02 (.03) | .02 (.03) | .02 (.03) | .96 |
| Null | Group characteristics do not explain ideology-prejudice association | $\hat{y}=0$ | .09 (.12) | .13 (.19) | .16 (.23) | .05 (.07) | .05 (.05) | – |

Association Test scores [18]). This addresses a key shortcoming of the original study. In particular, the original study used only feeling thermometers as an explicit measure of prejudice. Although feeling thermometers are common measures of prejudice (e.g., [19–21]), they are just one possible measure of group-based attitudes. The predictive accuracy of the models might be limited to feeling thermometers. If so, this would limit the utility of the predictive models to only studies that use feeling thermometers. This is potentially a substantial limitation given the diversity of existing prejudice measures. However, if the predictive models work well with alternative measures of prejudice, i.e., the explicit and reaction time measures, it suggests the model is much more generalizable. This would mean that this model could be used for different types of prejudice measures.

Further, the original models were built and tested using an absolute definition and operationalization of prejudice. That is, prejudice was defined as a negative group-based attitude (see also [22,23]). This means that a person who expresses negative attitudes about both men and women would be considered prejudiced against both men and women. Although this is a widely used definition and operationalization of prejudice, it is not the only definition. Another widely used definition and operationalization of prejudice is a relative definition. That is, prejudice is defined as a negative attitude about a group *relative* to one's attitudes about another group (see also [6,18,24]). This definition captures notions of bias and differential attitudes that are often associated with the concept of prejudice. Consider a person who expresses negative attitudes about both men and women. Under the relative definition of prejudice, they would not necessarily be considered prejudiced because there is no difference in their attitudes about men and women. In contrast, a person who expresses more negative attitudes about women relative to their attitudes about men would be considered prejudiced toward women. One important contribution of the current work is that we tested both operationalizations of prejudice.

The traditional IAT, for example, is a relative comparison between two groups. It assesses differences in participants' reaction times when associating groups with positive and negative words [18]. Similarly, explicit measures that ask respondents to choose which group they like best or look at the difference in negative feelings towards one group compared to another would also be relative measures of prejudice. Brandt's [1] models were not made for such relative measures. Therefore, in Study 2, when we tested relative measures, we built new models (conceptually based on those in Table 1) for relative measures of prejudice. After building models for relative prejudice, we tested the models on new data.

**The current studies**

Here, we tested four key models from Brandt [1], in addition to a null model (see Table 1), using a large dataset. Schmidt and colleagues [25] issued a call for registered reports and provided us with a new and very large (*N* = 261,119) dataset to conduct our analyses. This manuscript represents our Stage 2 findings. Studies 1 and 2 used the Ideology 2.0 dataset [25].

We replicated (Study 1) and extended (Study 2) Brandt's [1] work in two ways. First, in Study 1, we used models derived from the original work to estimate the extent to which the ideology-absolute-prejudice relationships for each group in the Ideology 2.0 data are explained by the models. If the original models are robust, they should predict the ideology-absolute-prejudice relationship in the Ideology 2.0 dataset with alternative explicit measures of absolute prejudice measures with some accuracy. If they are not predictive of the ideology-prejudice association, it will provide valuable insight into how ideology may relate to different types of prejudice and the boundary conditions of the models.

Study 2 went beyond this. In this study, we built new predictive models of *relative* prejudice. This contributes to the literature by adding conceptual and computational depth to existing work. We examined the predictive abilities of our models by leveraging the size of the Ideology 2.0 dataset to perform a train/test split [26]. The train/test method is a data analytic method that allows for model predictions by splitting a dataset into two parts. One part is the "training" set that is used to build the model, and the other part is the "test" set that is used to test and validate the model. Following the general approach taken by Brandt [1], we first trained new models using the perceived differences between target groups on perceptions of ideology, status, and choice to predict the association between participant ideology and relative prejudice toward the target groups (e.g., the difference in perceptions of Black and White people's perceived ideology predicting participants' ideology's association with Black vs. White relative prejudice). After estimating these models, we tested their predictive accuracy in the test set of data. We used these models to predict levels of relative prejudice toward each pair of target groups and compared the results of the models to the observed relationship between ideology and comparative prejudice in the data. This allowed us to generate new predictive models using a larger number of prejudice measures.

To perform the analyses in both studies, we needed to obtain estimates of how people perceive the target groups. These are the basis for the models built by Brandt [1] and are therefore necessary to estimate the models here. Specifically, we needed to know how the target groups are perceived in terms of their ideology, status, and choice. Some of the groups that we included in our models were not included in prior work, so it was unknown how they are perceived on these characteristics. We collected new data to obtain these perceptions, which we used for both Studies 1 and 2. These data allowed us to examine whether the models derived from Brandt [1] generalize to new sets of target groups using different measures of prejudice.

## Method

We first describe the details of the new data collection that we conducted to assess how various groups from the Ideology 2.0 dataset are perceived in terms of ideology, status, and choice. We then describe the details of Study 1 and Study 2.

### Ethics approval

We obtained ethics board approval to collect the new data as well as use Schmidt's archival data [25] (University of Virginia Institutional Review Board protocol number 2044847−1).

Participants in the Ideology 2.0 study participated in an online study. They read a consent document and advanced to the screen with the study instructions only if they agreed to participate. Because of the nature of online studies, it was not possible to obtain written consent as any signatures obtained would inevitably be linked to specific participants. The researchers who own the Ideology 2.0 dataset obtained approval from the University of Virginia's Institutional Review Board for this procedure. We accessed these data after receiving notification of in-principle acceptance (July 26, 2024). We did not have access to any information that could identify individual participants during or after data collection.

In the new target group perceptions data collection for this paper, the consent document preceded the survey. Participants read about the study and decided whether they would like to participate. Participants who elected to participate clicked an arrow button to advance to the survey (which indicated informed consent). If respondents opted not to participate, they were prompted to close out of the survey with no penalty. Our data collection did not involve collecting any identifying or sensitive information. Due to the nature of online surveys, it was not possible for participants to provide a signature on the consent document, as doing so would have made participant data identifiable given that the signature would be linked to their survey responses. The Institutional Review Board at Florida Atlantic University approved this procedure. All participants were recruited through Prolific on May 31, 2024 (we collected the data at this time to use them for another project).

### I. New data collection

**Participants and procedure.** We administered the survey via Prolific, which is an online survey website where participants can choose which studies to complete. Potential participants read a short description of the study and clicked on the survey link if they chose to participate in the study. Next, they read a consent form before proceeding with the survey. Participants filled out basic demographic information and then responded to the other survey items. We recruited a sample of 100 US citizens living in the US (approval rating > 95 on Prolific), with an equal number of men and women. The study took about 10 minutes and participants were paid $2.00 for their participation.

**Measures.** This new data collection was used to obtain group ratings for multiple registered reports using the Ideology 2.0 data. For the purposes of this study, we collected participants' perceptions of the target groups' ideological positions, status, and the extent to which membership in the target group is a choice (we used the measurements suggested by Brandt and Crawford [9] and used by Brandt [1]). Each participant rated each target group on each characteristic.

We included three groups in the new data collection (scientists, capitalists, and socialists) that were not part of our studies (as some of them did not have a pair or did not represent concrete groups). They were collected to conserve resources as other members of our research team needed them for another project. There is no reason to think that including these additional groups affected the group ratings. We calculated our sample size based on 100 ratings for 21 target groups (18 groups from the studies in this paper + 3 additional) which means we collected 2100 ratings in total. All survey items are contained in the S1 Appendix.

**Perceived ideology.** Ideological positions of the target groups were measured on a scale from 0 (*strongly liberal*) to 100 (*strongly conservative*). We paired the target groups with the following statement used by Brandt [1], "For each group, indicate whether you think the group is typically a liberal or conservative group."

**Perceptions of group status.** Status of target groups was measured on a scale from 0 (*low status*) to 100 (*high status*). Before asking about status of the target groups, we presented participants with this statement used by Brandt [1], "Some groups in society have higher status. That is, they have more education, they have more prestigious jobs, and they are more economically successful than other groups. Some groups have lower status. That is, they have less education, less prestigious jobs, and are less economically successful than other groups. And, of course, some groups are more in the middle."

**Perceived choice.** Choice was measured on a scale from 0 (*not at all*) to 100 (*very much*). Participants read this statement used by Brandt [1], "Sometimes people have choice and control of whether they belong to a particular group. Other times, they do not have much choice and control over whether they belong to a particular group." Then, they responded to the question, "To what extent can members of this group choose or control whether they actually belong to this group?" for each target group in the study.

**Demographics.** Additionally, we included demographic measures, including participants' own political orientations (1 = *strongly liberal*, 7 = *strongly conservative*). We also asked for age, gender, education level, income, and race/ethnicity using the same measures as were used in the Ideology 2.0 study.

## II. Studies 1 and 2: The Ideology 2.0 dataset

**Open access to data and code.** Here, we report how we determined sample size, how all participant exclusions were determined, and all manipulations and measures. All data used in this registered report and annotated R code (including MSE calculations, which are described in the project OSF Wiki) are on the project's OSF page: https://osf.io/buwp7/?view_only=3905d5d1d54b4a499483c03a089b9f6e. We pre-registered the studies and planned analyses before analyzing the data after in-principle acceptance of our registered report.

**Participants and procedure.** We used data from the Ideology 2.0 study for both Study 1 and 2. Data from the Ideology 2.0 study were collected between December 2007 and June 2012 from the Project Implicit website using a planned missingness design [25]. There were over 280,000 unique sessions, 40 reaction time measures (i.e., implicit measures), 30 self-report measures that matched the IAT targets, 25 individual difference questionnaires, and many individual self-report items. Participants were randomly assigned 15 minutes' worth of items. Participants either completed one reaction time measure and nine explicit measures on the same topic or completed reaction time and explicit measures for two different topics. Topics included groups (our focus), but also specific concepts (e.g., fascism) and specific people (e.g., George W. Bush).

Planned missingness is a strategy used in data collection where participants are randomly assigned to respond to only certain items [27,28]. The missing data points are missing completely at random (MCAR), so we can assume the missing data will not systematically impact the results. That is, while each participant has missing data for certain variables, there is no consistent pattern across participants regarding which variables are missing. Therefore, the missingness can effectively be ignored in the analyses [29]. Other researchers who have collected large datasets (e.g., [30,31]) have used MCAR data strategies to reduce the burden on participants and the cost of data collection.

The researchers from the Ideology 2.0 study made their exploratory data available to other researchers to use for registered report studies [25]. In the initial stage, Schmidt and colleagues released confirmatory *masked* data to researchers who requested it. From this confirmatory masked data, we determined that there are 24,296 relevant sessions for this study (we filtered out sessions where participants did not respond to our measures of interest and where respondents were not from the United States). Upon receipt of the full data, we confirmed the number of available participants and demographics. Table 2 contains participant demographics.

### Ideology 2.0 demographics measures used in Studies 1 and 2

The Ideology 2.0 dataset included these demographic measures: political ideology, age, education, ethnicity, religion, gender, and income. Liberal/conservative ideology was measured such that 1 = *very liberal* and 7 = *very conservative*. Participants self-reported their age in years. Education was measured with five categories ranging from no high school diploma to graduate-level education. Ethnicity had nine categories: American Indian/Alaska Native, East Asian, South Asian, Native Hawaiian or other Pacific Islander, White, Black or African American, and three categories describing some combination of these categories, or other. The religion variable initially contained 48 largely overlapping categories so we recoded it into Christian, Mormon, Jewish, Muslim, Hindu, Buddhist, and "other" categories. Gender had two categories, male and female. We included political ideology, age, education, ethnicity, religion, and gender in our analyses. We omitted income from our analyses because there was substantial missing data. Correlations between measures are included in Table 3.

### III. Study 1

**Analytic strategy – predicting absolute prejudice using existing models.** With the measures of absolute prejudice as the outcome in the Ideology 2.0 data, we used the same procedure for predicting ideology-prejudice associations Brandt [1] used, which involved using models that include perceived group characteristics to predict the ideology-prejudice

**Table 2. Demographics from the Ideology 2.0 Dataset.**

| Data Including IAT and Difference Score Explicit Measures | |
|---|---|
| N | 24296 |
| M age | 31.57 |
| SD age | 12.98 |
| % Men | 32.28 |
| % Women | 67.53 |
| % University degree | 46.22 |
| **Data Including Only Single Target Explicit Ratings** | |
| N | 6379 |
| M age | 31.65 |
| SD age | 12.98 |
| % Men | 33.27 |
| % Women | 66.53 |
| % University degree | 47.25 |

association for specific target groups. We used Brandt's [1] R code as a guide to reproduce the models (available online: https://osf.io/g28yc/). First, we identified target groups in the Ideology 2.0 data (18 in this case, with nine from Brandt's [1] original study and nine new groups). These groups are described in Table 4 below.

**Absolute prejudice measures.** To capture absolute prejudice levels toward each target group, we used four measures in the Ideology 2.0 dataset in Study 1. Participants responded to four measures of the valence of their actual feelings toward targets, their "gut" feelings about targets, and how positively and negatively they felt about targets. The names of the items in the dataset, the number of complete cases available for each item, item wording, and scale endpoints are listed in Table 5. The groups for which this is available, and which were included in analyses are shown in Table 4 above, with available cases for each.

**Power analyses for the models looking at the absolute measures.** We used the InteractionPowerR Shiny App for analytic power [32] to examine our ability to detect the relationship between perceived target characteristics and the ideology-prejudice relationship for the absolute measures. The app calculates power based on the anticipated correlations between predictors, predictors and the dependent variable, and the strength of the interaction to be tested. For the absolute measures with our final sample size ($N$ = 6,379), the analyses suggested we had a 90% chance of detecting an effect with correlations between our predictors as well as each predictor and the outcome slope in standardized units of .10, and an interaction term of .05. This suggests we were well-powered to detect small effects.

**Calculating predicted and observed prejudice values.** To examine whether Brandt's [1] models were robust, we input the values for perceptions of group ideology, status, and choice from our newly collected data into his equations (see Table 1 for the models and the S1 Appendix for the raw group characteristics values), with a separate equation for each target group. These equations give us the predicted ideology-prejudice association for each group. All the equations for the predicted values are in the code on the project's OSF page, and the OSF Wiki provides a guide as to where each component resides in the file. Then, we solved the predictive equations to generate estimates of the ideology-prejudice association for each target group in the context of each of the four models. We termed these estimates the *predicted* ideology-prejudice association. A fifth model, the null model (where the value of the association is 0), was also included for comparison purposes.

For example, referring to Table 1, to calculate the predicted ideology-prejudice association for the target group Black people using the ideology-only model (using Brandt's [1] original equation and the group ratings from our new data collection), we used this equation, $ŷ = 0.022–1.420(ideology)$, and substituted "(ideology)" with −0.1388:

**Table 3. Correlations between ideology 2.0 dataset variables.**

| | Age | Political ID | pref_xy | pos_x | pos_y | neg_x | neg_y | gut_x | gut_y | act_x | act_y | D_score | act_diff | gut_diff | neg_diff | pos_diff | D_score_xy |
|---|---|---|---|---|---|---|---|---|---|---|---|---|---|---|---|---|---|
| Age | – | -.07*** | .02*** | .02 | .04* | -.03 | -.01 | .00 | .01 | .00 | .01 | .02** | .00 | .01 | .01 | .01 | .02 |
| Political ID | | – | -.27*** | .17*** | -.12*** | .08*** | -.13*** | .21*** | -.22*** | .22*** | -.21*** | -.02** | -.28*** | -.27*** | -.16*** | -.22*** | -.17*** |
| pref_xy | | | – | -.43*** | .44*** | -.25*** | .46*** | -.60*** | .61*** | -.56*** | .60*** | .05*** | .76*** | .77*** | .57*** | .65*** | .49*** |
| pos_x | | | | – | .09*** | .18*** | -.36*** | .53*** | -.20*** | .55*** | -.15*** | -.06 | -.45*** | -.46*** | -.43*** | -.65*** | -.29*** |
| pos_y | | | | | – | -.25*** | .23*** | -.16*** | .54*** | -.13** | .54*** | -.04 | .46*** | .46*** | .37*** | .70*** | .13*** |
| neg_x | | | | | | – | .19*** | .40*** | -.15*** | .40*** | -.17*** | -.07* | -.37*** | -.35*** | -.60*** | -.32*** | -.16*** |
| neg_y | | | | | | | – | -.26*** | .49*** | -.26*** | .47*** | .01 | .49*** | .49*** | .67*** | .43*** | .24*** |
| gut_x | | | | | | | | – | -.22*** | .82*** | -.19*** | -.09*** | -.64*** | -.76*** | -.50*** | -.50*** | -.41*** |
| gut_y | | | | | | | | | – | -.19*** | .84*** | -.02 | .69*** | .81*** | .51*** | .56*** | .28*** |
| act_x | | | | | | | | | | – | -.17*** | -.10*** | -.73*** | -.62*** | -.50*** | -.50*** | -.39*** |
| act_y | | | | | | | | | | | – | -.02 | .79*** | .68*** | .50*** | .53*** | .25*** |
| D_score | | | | | | | | | | | | – | .05 | .05 | .06 | .01 | -.28*** |
| act_diff | | | | | | | | | | | | | – | .85*** | .67*** | .68*** | .45*** |
| gut_diff | | | | | | | | | | | | | | – | .66*** | .69*** | .48*** |
| neg_diff | | | | | | | | | | | | | | | – | .59*** | .35*** |
| pos_diff | | | | | | | | | | | | | | | | – | .35*** |
| D_score_xy | | | | | | | | | | | | | | | | | – |

Table 3 contains the correlations between centered age, rescaled political orientation, and the rescaled prejudice measures that were included in the analyses. Significant correlations are denoted by asterisks (* $p < .05$, ** $p < .01$, *** $p < .001$).

**Table 4. Number of absolute measure responses per distinct group in the data.**

| Groups from current study | In Brandt (2017) Study? | Number of Participants Available |
|---|---|---|
| **Absolute Measures** | | |
| Gay | Yes | 731 |
| Straight | **No** | 731 |
| Democrats | Yes | 757 |
| Republicans | Yes | 757 |
| Liberals | Yes | 738 |
| Conservatives | Yes | 738 |
| Non-Profits | **No** | 690 |
| Corporations | **No** | 690 |
| Labor | **No** | 644 |
| Management | **No** | 644 |
| Foreign | **No** | 692 |
| Local | **No** | 692 |
| Black | Yes | 731 |
| White | Yes | 731 |
| Religious | Yes | 768 |
| Atheist | Yes | 768 |
| Mother | **No** | 738 |
| Father | **No** | 738 |

**Table 5. Absolute measures of prejudice in the ideology 2.0 dataset.**

| Item | Wording | Scale endpoints | N | % Responded |
|---|---|---|---|---|
| **Items used to rate standalone targets (Sample 2)** | | | | |
| gut_x | What are your gut feelings toward x? | 7 = Strongly positive, 6 = Moderately positive, 5 = Slightly positive, 4 = Neither positive nor negative, 3 = Slightly negative, 2 = Moderately negative, 1 = Strongly negative | 3597 | 56.39 |
| gut_y | What are your gut feelings toward y? | 7 = Strongly positive, 6 = Moderately positive, 5 = Slightly positive, 4 = Neither positive nor negative, 3 = Slightly negative, 2 = Moderately negative, 1 = Strongly negative | 3597 | 56.39 |
| act_x | What are your actual feelings toward x? | 7 = Strongly positive, 6 = Moderately positive, 5 = Slightly positive, 4 = Neither positive nor negative, 3 = Slightly negative, 2 = Moderately negative, 1 = Strongly negative | 3596 | 56.37 |
| act_y | What are your actual feelings toward y? | 7 = Strongly positive, 6 = Moderately positive, 5 = Slightly positive, 4 = Neither positive nor negative, 3 = Slightly negative, 2 = Moderately negative, 1 = Strongly negative | 3596 | 56.37 |
| neg_x | Considering only the negative things about x and ignoring the positive things, how negative are those things? | 1 = Extremely negative, 2 = Very negative, 3 = Moderately negative, 4 = Slightly negative, 5 = Barely negative, 6 = not at all negative | 3627 | 56.86 |
| neg_y | Considering only the negative things about y and ignoring the positive things, how negative are those things? | 1 = Extremely negative, 2 = Very negative, 3 = Moderately negative, 4 = Slightly negative, 5 = Barely negative, 6 = not at all negative | 3629 | 56.89 |
| pos_x | Considering only the positive things about x and ignoring the negative things, how positive are those things? | 6 = Extremely positive, 5 = Very positive, 4 = Moderately positive, 3 = Slightly positive, 2 = Barely positive, 1 = not at all positive | 3628 | 56.87 |
| pos_y | Considering only the positive things about y and ignoring the negative things, how positive are those things? | 6 = Extremely positive, 5 = Very positive, 4 = Moderately positive, 3 = Slightly positive, 2 = Barely positive, 1 = not at all positive | 3633 | 56.95 |

Table 5 contains the names of the items in the dataset, the number of complete cases available for each item, item wording, and scale endpoints for the absolute measures of prejudice.

ŷ = 0.022–1.420(−0.1388). Solving for y, we arrive at the estimate of the ideology-prejudice association, .22. This means for that any group with a perceived ideology of −0.1388 (in our sample of target groups, Black people; the value was midpoint-centered), the model predicts that the association between ideology and prejudice is .22. The interpretation of this estimate depends on how ideology and prejudice is coded. We code participant ideology to range from 0 to 1 (0 = very *liberal* to 1 = very *conservative*), and prejudice is coded to range from 0 to 1 (0 = *low levels of prejudice* to 1 = *high levels of prejudice*). This means that the model predicts that as we go from the lowest point on the ideology scale (0 = *very liberal*) to the highest (1 = *very conservative*), participants are predicted to score higher on the prejudice measure by .22 (or about 22% of the scale range).

To obtain the *observed* values, we regressed each measure of prejudice for each target group on ideology and demographic control variables [33]. In these analyses, we used American Indian/Alaskan Native as the reference category for ethnicity, no high-school diploma for the education reference category, Christian as the religion reference category, male as the gender reference category, and mean-centered age. The measures of prejudice and ideology were rescaled to range from 0 to 1. We used the same demographic controls as Brandt [1], except for income, which we excluded because of large amounts of missing data. The estimate for ideology from this model was our observed ideology-prejudice estimate for each target group and each measure of prejudice.

Once we obtained the predicted and observed values, we examined how well the predicted association from each of our four models mapped onto the observed association for each target group in the Ideology 2.0 data. To do this, we estimated the Mean Squared Error (MSE) of the observed ideology-prejudice association compared to the predicted association. MSEs serve as a means of evaluating the accuracy of a predictive model by comparing predicted and actual values. To calculate MSEs, we subtracted the actual ratings of prejudice across groups from the estimates from the equations, squared these values, and then found the average. Then, we ran a mixed ANOVA and post-hoc tests using the jmv package in R [34] to determine whether the differences between model MSEs were statistically significant ($a$ = .05). We opted to use the original *p* values for the post-hoc tests as opposed to using a procedure like Bonferroni or Tukey corrections because we did not wish to omit potentially meaningful significant results based on stringent adjustments [35]. The model with the lowest MSE (and the difference between that model and the next-lowest model was statistically significant) was considered the most accurate predictive model of prejudice in this context. The calculations for the MSEs and the mixed ANOVA with post-hoc tests are included in the code on the project's OSF page. The OSF Wiki details the lines of the code correspond to each calculation and test.

**Comparing measure performance.** To examine measure performance, we looked at how well the predictive estimates for each model mapped onto the analyses for each absolute measure of prejudice. We also compared how well each model performed compared to the other models. For our analyses, we first calculated the MSEs for all absolute prejudice measures across all groups for each model. Then, we calculated MSEs for each absolute measure separately for each model. We estimated a mixed ANOVA with each of the five types of prejudice measures (all measures in combination, and each of the four individual prejudice measures) as a within-subjects factor and model type as a between-subjects factor with five levels to test if model accuracy depends on the models and type of measures.

## Results

For these analyses, we examined the effects of model, the effect of type of prejudice measure, and their interaction for all 18 target groups together. Using an alpha level of .05, the effect of type of prejudice measure was significant, $F(4,340)$ = 16.69, $p < .001$. This indicates the predictive equations were more accurate at predicting the ideology-prejudice association for some measures of prejudice than others. There was a significant effect of model type, $F(4,85)$ = 3.08, $p = .020$. This indicates some models had greater predictive accuracy than others. Additionally, there was a significant interaction between outcome measure and model type, $F(16,340)$ = 2.70, $p < .001$. This indicates that the extent of a model's accuracy was dependent on the outcome measure.

The mean differences for the main effect of the type of prejudice measure are reported in Table 6. The average of all measures was better at predicting the ideology-prejudice association than the single measures for actual and gut prejudice. The actual prejudice measure performed better than the gut measure. The positive prejudice measure performed better than the average of all measures and better than the single actual and gut feeling measures. The negative measure of prejudice performed better than the average of all measures as well as the single actual and gut feeling measures. There was no significant difference between the positive and negative measures. In short, when averaging across all models, the models were most accurate for the positive and negative measures and the least accurate for the gut measure. The mean differences for the main effect of model type are reported in Table 7. Overall, the ideology-only model showed the best performance, as it had a significantly lower MSE than all other models except for the combined ideology, status, and choice model. Further, the ideology-only model was more parsimonious than the model including all three group attributes (i.e., the ideology, status, and choice model). This replicates the findings of Brandt [1], who also used MSEs and *p*-values to compare model performance. However, considering statistical significance alone, the ideology-only and ideology, status, and choice models were tied in terms of accuracy. The ideology, status, and choice model performed better than the status-only, choice-only, and null models. The status-only and choice-only models did not significantly differ from the null model, nor from each other. Additionally, there was a significant interaction between model and measure type (the post hoc comparisons are included in the S2 Appendix). The MSE for each model and measure type are in Table 1. Examining this table reveals several patterns (confirmed in the S2 Appendix). First, the ideology-only and the ideology, status, and choice models had low MSEs across all measure types (MSEs ranged [.01, .02]). Second, across all measure types, the ideology-only and the ideology, status, and choice models were the most accurate models. Third, the status-only, choice-only, and null models all had much more variation in predictive accuracy depending on measure type (status-only MSEs range [.04, .14], choice-only MSEs range [.05, .16], null MSEs range [.05, .16]). In combination, this shows that the ideology-only and ideology, status, and choice models perform well across four different outcome variables. Fig 1 depicts the predicted and observed slopes for the absolute measures per measure type.

For the curious reader, we also ran the Study 1 analyses without the demographic control variables and found the same pattern of results (see S3 Appendix). As our Stage 1 report was accepted with the inclusion of controls in our analysis plan, we do not interpret those results further in-text.

**Exploratory analyses for explicitly political target groups.** Although the patterns in our planned analyses replicated Brandt [1], we explored whether our models were more predictive of the ideology-prejudice association for

**Table 6. Post hoc tests for absolute prejudice measures.**

| Comparison | | | | | |
|---|---|---|---|---|---|
| Prejudice Measures | Mean Difference | SE | df | t | p |
| All Measures – Actual | −0.03 | 0.01 | 85 | −4.57 | <.001 |
| All Measures – Gut | −0.05 | 0.01 | 85 | −4.83 | <.001 |
| All Measures – Positive | 0.02 | 0.01 | 85 | 2.98 | .004 |
| All Measures – Negative | 0.02 | 0.01 | 85 | 2.85 | .006 |
| Actual – Gut | −0.02 | 0.004 | 85 | −4.84 | <.001 |
| Actual – Positive | 0.04 | 0.01 | 85 | 3.95 | <.001 |
| Actual – Negative | 0.05 | 0.01 | 85 | 3.76 | <.001 |
| Gut – Positive | 0.06 | 0.01 | 85 | 4.28 | <.001 |
| Gut – Negative | 0.06 | 0.02 | 85 | 4.13 | <.001 |
| Positive – Negative | 0.003 | 0.004 | 85 | 0.67 | .507 |

Table 6 contains the mean differences for the main effect of the type of absolute prejudice measure.

**Table 7. Post hoc tests for model comparisons.**

| Comparison | | | | | |
| Model | Mean Difference | SE | df | t | p |
|---|---|---|---|---|---|
| Ideology − Status | −0.07 | 0.03 | 85 | −2.12 | .037 |
| Ideology − Choice | −0.08 | 0.03 | 85 | −2.36 | .021 |
| Ideology − Ideology + Status + Choice | −0.003 | 0.03 | 85 | −0.09 | .930 |
| Ideology − Null | −0.08 | 0.03 | 85 | −2.42 | .017 |
| Status − Choice | −0.01 | 0.03 | 85 | −0.24 | .810 |
| Status − Ideology + Status + Choice | 0.07 | 0.03 | 85 | 2.03 | .045 |
| Status − Null | −0.01 | 0.03 | 85 | −0.31 | .760 |
| Choice − Ideology + Status + Choice | 0.08 | 0.03 | 85 | 2.27 | .026 |
| Choice − Null | −0.002 | 0.03 | 85 | −0.06 | .948 |
| Ideology + Status + Choice − Null | −0.08 | 0.03 | 85 | −2.34 | .022 |

Table 7 contains the mean differences for the main effect of model type.

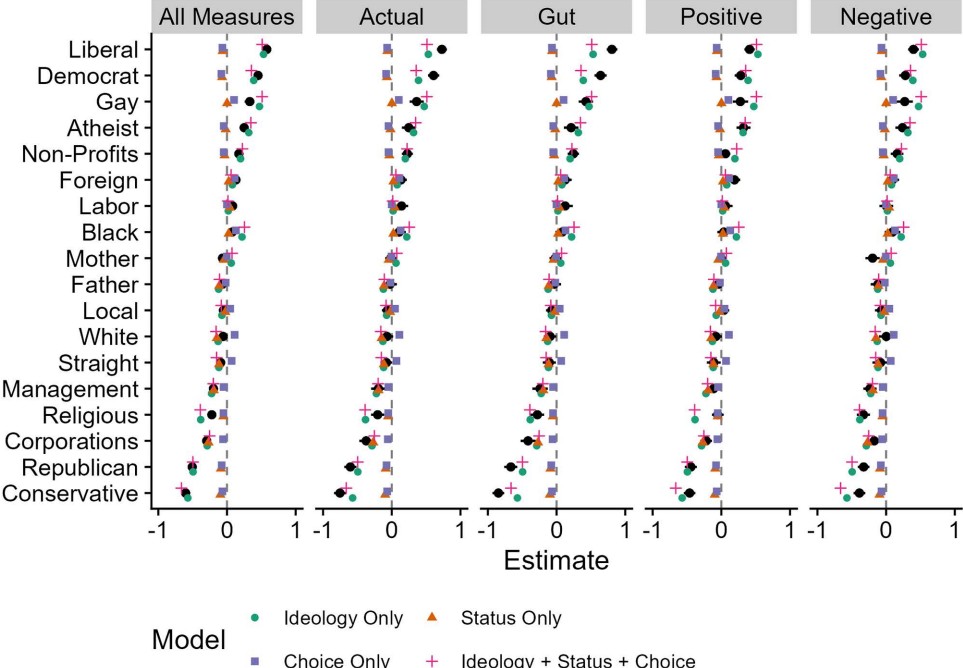

**Fig 1. Depiction of actual and model estimated slopes per measure type.**

explicitly political groups. This seemed relevant to us given the results of the relative measures models reported below (i.e., results were less strong in the subset of measures that did not include explicitly political groups). While many of the included groups can be considered political to some extent, some groups are explicitly political and tied to liberal-conservative politics in the United States (i.e., liberals, conservatives, Democrats, Republicans).

To examine the impact of target group on the analyses, we also ran the same analysis focusing on only the most explicitly political groups (Democrat, Republican, Liberal, and Conservative) and found significant effects for measure type ($F$(4,

60) = 129.41, $p < .001$), model type ($F(4, 15) = 17.68$, $p < .001$), and the interaction between model and measure ($F(16, 60) = 16.35$, $p < .001$). This largely replicates the analyses using all of the groups.

When we ran the analyses on the remaining target groups, there was no significant effect of measure type ($F(4, 260) = 1.63$, $p = .158$), model type ($F(4, 65) = 2.02$, $p = .103$), nor a significant interaction ($F(4, 260) = 1.63$, $p = .061$) between model and measure type. This suggests that the models (including the null model) perform equally well (or equally poorly) when examining non-ideological groups. We did not propose this analysis at Stage 1, so we do not interpret it strongly here. However, the analysis suggests that it is possible that the groups that best distinguish between the predictive models are explicitly political groups. We further discuss this in the General Discussion considering the results of Study 2. See the S4 Appendix for the exploratory analysis results.

## Study 1 discussion

In our Stage 1 report, we predicted that the ideology-only model would be the best performer. The ideology model was a strong performer, but as in the case of Brandt's [1] original study, there were no significant differences between the predictive ability of the ideology-only and ideology, status, and choice models. Additionally, our results closely mirrored Brandt's [1] findings such that the status-only and choice-only models had weaker performances and were not significantly different from the null model. The mirroring of results from Brandt [1] goes beyond the pattern of significance. The MSEs of the ideology-only and ideology, status, and choice models are also similar in our study (MSEs ranged [.01, .02]) and Brandt's study (MSEs ranged [.01, .04]). This successful replication indicates Brandt's [1] models are robust across different types of absolute measures and across different samples. Although the ideology-only and ideology, status, and choice models were not significantly different from each other, we suggest the ideology model is the best choice due to its simplicity.

We want to acknowledge that the results of our exploratory analyses with explicitly political groups removed suggested that the strength of the ideology-only and ideology, status, and choice models may be attributed to the inclusion of these explicitly political groups. Although we remain cautious in over-interpreting these exploratory analyses, as we chose to do those analyses *post-hoc* after our Stage 1 manuscript was accepted, we want to discuss two plausible ways to interpret these results. First, these exploratory results could reveal a major limitation of the ideology-only model in that the model only works well when these explicitly political groups are included. Second, these exploratory results could reveal that to fully test and understand models of ideology and prejudice, it is important to use a wide range of groups that represent the broader population of groups in society. For many groups, the models make similar predictions (see the black dots in Fig 1), but their predictive advantages only emerge when including those groups where they make different predictions. To test between models, it is necessary to include groups where the models make different predictions.

## IV. Study 2

**Analytic strategy – creating new models to predict comparative prejudice.** In Study 2, our goal was to examine which characteristics of target groups were most predictive of the comparative ideology-prejudice relationship. To do this, we first split our data into two parts (80/20 split), a training set (80%) and a test set (20%, e.g., [11,36]), at random for each target group pair [26]. Then, using Brandt's [1] model-building technique, we estimated new predictive models using the lme4 package in R [33], using the training portion of the data. These models gave us our *predicted* ideology-prejudice association values. Using the test portion of the data, we estimated multilevel models that estimated the ideology-prejudice relationship using the *differences* between the relevant target pair characteristics. These models included four conceptually identical models to those used for the ideology-absolute prejudice models (in Study 1), which provided us with our *observed* ideology-prejudice association values. The train-test method allowed us to do three things: incorporate a prediction model that was designed specifically for our research questions [26], measure prediction accuracy using a new sample, and assess overall ideological-prejudice association prediction accuracy. The code for the models used to

obtain the predicted and observed values are included in the code on the project's OSF page. The OSF Wiki contains details related to which lines of the code correspond to each calculation and test.

In the ideology-only model, we included our measure of the difference in perceived ideology for the groups in the pairs predicting how well participant ideology explains the difference in prejudice between each target in a pair (e.g., Black vs. White people). Additional models used perceived differences in status and choice, respectively to predict the relationship between ideology and comparative prejudice for each group, and a fourth model used perceived differences in ideology, status, and choice in combination to predict the relationship between participant ideology and comparative prejudice. The fifth model, the null model, assumes the ideology-prejudice association will be 0. This model was included for comparison purposes.

**Explicit relative prejudice measures.** We included five measures of explicit relative prejudice in the outcome variable of our predictive and observed models. First, participants were asked to indicate which group they preferred when given a choice between two groups (e.g., Black people and White people). Then, using the "gut," "actual," "positive," and "negative" feelings measures described above, difference scores were created for each respective measure to determine preferences for some target groups over others. The "gut difference" measure was calculated by subtracting the gut feeling score for group X minus the score for group Y (e.g., where White people are group Y and Black people are group X). The "actual difference" measure was calculated by subtracting the actual feeling score for group X minus the score for group Y. The "positive difference" measure was calculated by subtracting the positive feeling score for group X minus the score for group Y. The "negative difference" measure was calculated by subtracting the negative feeling score for group X minus the score for group Y. In all cases except one, the more traditionally liberal group is coded as group X (e.g., gay people). The one case where the more traditionally conservative group is coded as group X is the religious people vs. atheists pairing. In this case, we reversed the score for this group comparison. The number of participants who responded to each measure is included in Table 8. The groups for which this is available, and which were included in the analyses are shown in Table 9 with available cases for each.

**Reaction time outcomes.** We also included reaction time variables in our outcome variable. When completing the IAT [18], participants pair groups (i.e., Black and White) with positive and negative stimuli (i.e., "good" and "bad" words). The outcome measure of the IAT is a reaction time difference score, called the D-score [18]. This score indicates a preference for one group over another, such that a faster reaction time when pairing one group with positive stimuli (as opposed to negative stimuli) indicates a preference for that group. In this dataset, D-scores are coded in the same way as the differences between explicit measures (see the Target X and Target Y columns in Table 10). We used the IAT as a measure of comparative prejudice. The number of participants who completed IAT tasks is listed in Table 8 above. We did not include data for three pairs (Democrat/Republican, Liberal/Conservative, and Religious/Atheist) because the IAT task for these groups involved comparing the target group to the self. In our analyses, we used both the explicit

**Table 8. Measures of explicit relative prejudice and IAT D-score in the ideology 2.0 Dataset.**

**Items used in the data including IAT responses (Sample 1)**

| Item | Wording | N | % Responded |
|------|---------|---|-------------|
| pref_xy | Which group do you prefer, x or y | 24249 | 99.81 |
| gut_diff | Gut feelings toward x – gut Feelings toward y | 3594 | 14.79 |
| act_diff | Actual feelings toward x – actual feelings Toward y | 3594 | 14.79 |
| neg_diff | Negative feelings toward x – negative feelings toward y | 3620 | 14.90 |
| pos_diff | Positive feelings toward x – positive feelings toward y. | 3622 | 14.90 |
| D_score_xy | IAT D-scores in same direction as explicit measure scoring | 9643 | 39.69 |

Table 8 contains the number of participants who responded to each type of explicit relative prejudice measure. The last row represents the IAT D-score.

**Table 9. Number of relative measure responses per distinct group in the data.**

| Groups from current study | In Brandt (2017) Study? | Number of Participants Available |
|---|---|---|
| *Relative Measures* | | |
| Black vs. White | Yes | 2748 |
| Gay vs. Straight | **Straight only** | 2869 |
| Mother vs. Father | **No** | 2777 |
| Foreign vs. Local | **No** | 2659 |
| Labor vs. Management | **No** | 2543 |
| Non-Profits vs. Corporations | Yes | 2608 |
| Democrat vs. Republican | Yes | 2631 |
| Liberal vs. Conservative | Yes | 2634 |
| Religious vs. Atheist | Yes | 2827 |

**Table 10. IAT tasks.**

| Target X | Target Y | Preference IAT Available |
|---|---|---|
| Gay | Straight | Evaluation (Good/ Bad) |
| Non-profits | Corporations | Evaluation (Good/ Bad) |
| Labor | Management | Evaluation (Good/ Bad) |
| Foreign | Local | Evaluation (Good/ Bad) |
| Black | White | Evaluation (Good/ Bad) |
| Mother | Father | Evaluation (Good/ Bad) |

We did not use reaction time measures from IATs comparing a group to the self (i.e., Democrat/Republican, Liberal/Conservative, and Religious/Atheist).

relative measures and the reaction time outcomes because they are measures of comparative prejudice (they were also moderately correlated, see Table 3). Specifically, these measures both involve making a choice to indicate preference for one group over another. In the context of our multilevel models, each measure was nested within participants in separate rows.

**Power analyses for the comparative measures.** We used the InteractionPowerR Shiny App for analytic power [32] to examine our ability to detect the relationship between perceived target characteristics and the ideology-prejudice relationship. For the comparative measures using our final sample size ($N = 24{,}296$), the analyses suggested we had an 82% chance of detecting an effect with correlations between our predictors as well as each predictor and the outcome slope in standardized units of .06, and an interaction term of .02. This suggests we were well-powered to detect small effects.

**Training the data.** We used the training data set to create our predictive models of comparative prejudice. To examine measures of comparative prejudice against target groups we estimated multilevel models using the lme4 package in R [33] on the first random 80% of the Ideology 2.0 data. In the models, we nested the measures of comparative prejudice against different target group pairs within participants and included random slopes to account for overall differences between participants and target pairs in terms of overall levels of comparative prejudice. For each respective model, we included perceived differences in ideology, status, choice, and all three measures simultaneously as predictors of the ideology-comparative prejudice relationship. These were treated as random slopes [37]. This allowed us to examine the amount that each of these perceived traits impacts the relationship between participant ideology and comparative prejudice.

Additionally, we controlled for gender with men as the reference group, education with less than a high school education as the reference group, religion with Christians as the reference group, ethnicity with American Indian/Alaska Native as the reference group and mean-centered age. All variables, except the target characteristic differences, were recoded to range from 0–1 so that the coefficients could be interpreted as the percent of change in the outcome as one goes from the lowest to the highest value in the measure. We used the same control variables as Brandt [1] except for income, which we excluded because of large amounts of missing data. To create the target characteristic difference values, we rescaled the original target characteristic values from 0–1 and then subtracted the traditionally higher-status group's value from the lower-status group's value for each pair (e.g., Gay ideology value – Straight ideology value). We used this method because recoding the difference score to range from 0–1 would result in a dependent measure indicating difference versus no difference, but in a non-directional fashion, which would make interpretation difficult.

As an example of how these models can be used to predict the ideology-prejudice association for relative measures, here is our status-only model: $\hat{y} = -0.480 - 1.016(\text{status})$. If we wanted to estimate the ideology-prejudice association for the religious/atheist pair, we would input the difference in status rating between religious people and atheists (calculated by subtracting atheist – religious), −0.0403. Then, we would insert the status value into the equation, $\hat{y} = -0.480 - 1.016$ (−0.0403) to arrive at the predicted value, −.44. The new predictive models are included in Table 11.

**Testing model fit.** After we trained our models using the training data, we used the test data (the remaining 20% of the Ideology 2.0 dataset, for each target group) to estimate the observed relationship between ideology and comparative

**Table 11. Predictive equations generated from the training data.**

| Model Name | Theoretical Implication | Model | MSE (SD) Estimate Across Outcomes | MSE (SD) for Actual Prejudice Difference Outcome | MSE (SD) for Gut Prejudice Difference Outcome | MSE (SD) for Positive Prejudice Difference Outcome | MSE (SD) for Negative Prejudice Difference Outcome | MSE (SD) for Preference Difference Outcome | MSE (SD) for D-Score Difference Outcome | Correlation Between Predicted and Observed Values (*r*) |
|---|---|---|---|---|---|---|---|---|---|---|
| ideology-only | Ideological differences explain ideology-prejudice association | $\hat{y} = 0.080 + 1.070$ (ideology) | .01 (.01) | .02 (.05) | .02 (.02) | .04 (.05) | .04 (.05) | .02 (.03) | .02 (.03) | .92 |
| status-only | Status differences explain ideology-prejudice association | $\hat{y} = -0.480 - 1.016$ (status) | .06 (.06) | .08 (.06) | .07 (.08) | .03 (.03) | .03 (.044) | .09 (.11) | .05 (.06) | .43 |
| choice-only | Choice differences explain ideology-prejudice association | $\hat{y} = -0.394 - 1.738$ (choice) | .05 (.05) | .05 (.06) | .05 (.08) | .03 (.04) | .03 (.05) | .07 (.09) | .04 (.04) | .62 |
| ideology, status, and choice | A combination of group characteristics explains ideology-prejudice association | $\hat{y} = 0.132 + 1.259$ (ideology) $-0.058$(status) + 0.526(choice) | .01 (.02) | .03 (.05) | .02 (.02) | .05 (.08) | .05 (.07) | .02 (.02) | .02 (.02) | .91 |
| null | Group characteristics do not explain ideology-prejudice association | $\hat{y} = 0$ | .16 (.25) | .14 (.22) | .17 (.26) | .04 (.04) | .05 (.06) | .23 (.33) | .01 (.01) | – |

Table 11 includes our newly built equations based on the training data. The MSEs and SDs for each model and for each type of measure are also presented in the table.

prejudice for each target group pair. We regressed each measure of relative prejudice for each target group on ideology and demographic control variables.

After obtaining the observed values, we determined which predictive model captures observed comparative prejudice most closely. We did this by estimating the MSE of the observed ideology-comparative prejudice association compared to the predicted association. By comparing the MSEs of each of the five models predicting the participant ideology-prejudice relationship across targets and the observed values of prejudice against each target pair in the test set, we were able to test which model(s) were most effective in predicting prejudice across targets. Lower MSEs when comparing the slopes in the estimates to the observed levels of participant ideology-comparative prejudice indicated better fit in this context. The MSEs for each model and measure are included in Table 11 above.

## Results

Our first task in this study was to build the predictive equations based on the training data. The equations are listed in Table 11 above, along with the MSEs for each model and measure.

After building the models and obtaining the predicted and observed values, we calculated the MSEs that represented the comparison between the predicted and observed ideology-prejudice association values for each target pair, for each model and measure type. We estimated a mixed ANOVA with each of the seven types of prejudice measures (all measures in combination, and each of the six relative prejudice measures) as a within-subjects factor and model type as a between-subjects factor with five levels to test if model accuracy depends on the models and type of measures.

Using an alpha level of .05, the effect of prejudice measure type was not significant, $F(6, 150) = 1.32$, $p = .251$. This indicates that the predictive equations were equally accurate in predicting prejudice across all measures. The effect of model type was not significant, $F(4, 25) = 0.95$, $p = .451$. This indicates that there was no significant difference in predictive accuracy between the models. The mean differences for the main effect of the type of prejudice measure are reported in Table 12, and the mean differences for the main effect of model type are reported in Table 13. Two of the measure type pairwise comparisons were significant, but none of the model type pairwise comparisons were significant. Additionally, the interaction between outcome measure and model type was not significant, $F(24, 150) = 1.17$, $p = .282$ (the code to run the post hoc analyses is included in the Study 2 R Markdown file on our OSF page). This indicates that whether a model was accurate or not did not depend on the outcome measure. To augment our usage of MSEs and $p$-values to compare model performance, we also ran correlations between the predicted and observed values (see Table 1 and Table 11). We found that the ideology-only and ideology, status, and choice models had the highest correlations between predicted and observed values, with values nearly as high as the models in Study 1. Given that we stated we would use $p$-values and MSEs in our Stage 1 report, we interpret those metrics in this work. Fig 2 depicts the predicted and observed slopes for the relative measures per measure type.

As we did in Study 1, we also ran the Study 2 analyses without the demographic control variables and found the same pattern of results (see S3 Appendix). We interpreted the analyses with the controls here, as this is consistent with our Stage 1 manuscript, and our intention in Study 2 is to run analyses that parallel Study 1.

### Exploratory analyses with D-score removed

Study 2 differed from Study 1 because there were fewer observations (9 *pairs* of groups instead of 18 *individual* groups). Further, for the D-Score measure, there were only 6 pairs because the IAT measures for Democrats/Republicans, Liberals/Conservatives, and Religious/Atheist asked participants to compare these groups to *themselves* instead of good/bad evaluations. This significantly reduced the number of available pairs for our models, as well as removed explicitly political groups. To examine the impact of this reduced number of groups, and explicitly political groups, we ran a set of exploratory analyses. Specifically, we conducted the Study 2 analyses again with the D-Score data removed, meaning we were able to predict the ideology/prejudice association for all groups, including explicitly political groups. Importantly,

**Table 12. Post hoc tests for relative prejudice measures.**

| Comparison | | | | | |
|---|---|---|---|---|---|
| **Prejudice Measures** | **Mean Difference** | **SE** | **df** | **t** | **p** |
| All Measures – Actual | −0.01 | 0.005 | 25 | −1.88 | .071 |
| All Measures – Gut | −0.02 | 0.01 | 25 | −2.75 | .011 |
| All Measures – Positive | −0.002 | 0.004 | 25 | −0.38 | .706 |
| All Measures – Negative | −0.01 | 0.004 | 25 | −1.44 | .163 |
| All Measures – Preference | −0.01 | 0.01 | 25 | −2.09 | .047 |
| All Measures – D-Score | −0.004 | 0.005 | 25 | −0.82 | .418 |
| Actual – Gut | −0.01 | 0.01 | 25 | −0.84 | .408 |
| Actual – Positive | 0.01 | 0.01 | 25 | 0.90 | .376 |
| Actual – Negative | 0.002 | 0.01 | 25 | 0.32 | .749 |
| Actual – Preference | −0.002 | 0.01 | 25 | −0.31 | .764 |
| Actual – D-Score | 0.005 | 0.01 | 25 | 0.67 | .506 |
| Gut – Positive | 0.01 | 0.01 | 25 | 1.90 | .069 |
| Gut – Negative | 0.01 | 0.01 | 25 | 1.14 | .261 |
| Gut – Preference | 0.004 | 0.005 | 25 | 0.89 | .381 |
| Gut – D-Score | 0.01 | 0.01 | 25 | 1.32 | .198 |
| Positive – Negative | −0.005 | 0.003 | 25 | −1.64 | .112 |
| Positive – Preference | −0.01 | 0.01 | 25 | −1.09 | .283 |
| Positive – D-Score | −0.002 | 0.01 | 25 | −0.47 | .642 |
| Negative – Preference | −0.004 | 0.01 | 25 | −0.49 | .629 |
| Negative – D-Score | 0.002 | 0.004 | 25 | 0.61 | .547 |
| Preference – D-Score | 0.01 | 0.01 | 25 | 0.71 | .484 |

Table 12 contains the mean differences for the main effect of the type of relative prejudice measure.

**Table 13. Post hoc tests for model comparisons.**

| Comparison | | | | | |
|---|---|---|---|---|---|
| **Model** | **Mean Difference** | **SE** | **df** | **t** | **p** |
| Ideology – Status | −0.03 | 0.02 | 25 | −1.65 | .110 |
| Ideology – Choice | −0.02 | 0.02 | 25 | −0.98 | .337 |
| Ideology – Ideology + Status + Choice | −0.001 | 0.02 | 25 | −0.06 | .950 |
| Ideology – Null | −0.02 | 0.02 | 25 | −0.83 | .416 |
| Status – Choice | −0.01 | 0.02 | 25 | 0.68 | .504 |
| Status – Ideology + Status + Choice | 0.03 | 0.02 | 25 | 1.59 | .124 |
| Status – Null | 0.02 | 0.02 | 25 | 0.83 | .415 |
| Choice – Ideology + Status + Choice | 0.02 | 0.02 | 25 | 0.926 | .369 |
| Choice – Null | 0.003 | 0.02 | 25 | 0.15 | .881 |
| Ideology + Status + Choice – Null | −0.02 | 0.02 | 25 | −0.768 | .4 |

Table 13 contains the mean differences for the main effect of model type.

this analysis is exploratory and was not described in our accepted Stage 1 manuscript. However, we felt it was an important set of analyses to consider, given the groups that were omitted and the difference from the Study 1 results. We do not make strong claims about the results of these analyses, given that they were not pre-registered or a part of the initial Stage 1 manuscript.

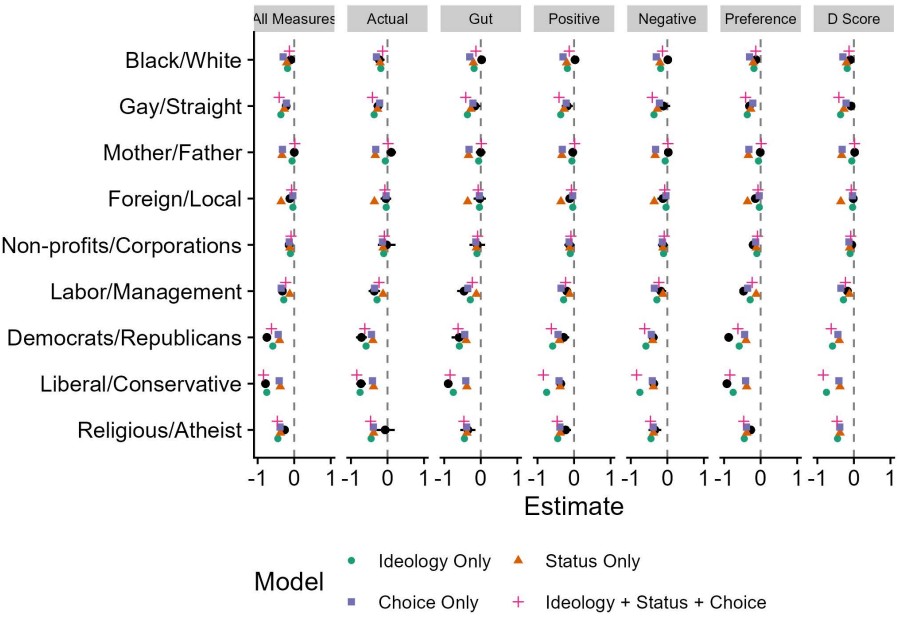

**Fig 2. Depiction of actual and model estimated slopes for relative measures.**

With the D-Score data removed and using an alpha level of .05, the effect of prejudice measure type was significant, $F(5, 200) = 3.10$, $p = .010$. This indicates that the predictive equations were more accurate at predicting the ideology-prejudice association for some measures of explicit prejudice than others. The effect of model type was not significant, $F(4, 40) = 2.07$, $p = .103$. This indicates that there was no significant difference in predictive accuracy between the models. Importantly, the interaction between outcome measure and model type was significant, $F(20, 200) = 2.26$, $p = .002$. This indicates that the extent of a model's accuracy was dependent on the outcome measure.

The mean differences for the main effect of the type of prejudice measure are reported in Table 14. All measures together, and the positive and negative measures, all had greater predictive accuracy than the preference measure. The mean differences for the main effect of model type are reported in Table 15. Two of the model type pairwise comparisons were significant (the respective differences between the null model and the ideology-only and ideology, status, and choice models). There was a significant interaction between model and measure type, although we do not include the table here because of its considerable size. For interested readers, the code to run the post hoc analyses is included in the Study 2 R Markdown file on our OSF page. All the significant post hoc interactions involved comparisons with the null model, indicating that the other four models were stronger predictors than the null model across a variety of measures (especially the preference measure).

### Exploratory analyses with explicit and implicit measures separated

Per a reviewer's suggestion, we also ran the Study 2 analyses with the explicit and implicit relative measures separated. Although the explicit and implicit measures were moderately correlated (ranging from .35 −.45, see Table 3), these correlations indicate potential differences between the measures. For full results, see the S5 Appendix. Most importantly, the effect of model type was not significant, $F(4, 40) = 1.43$, $p = .243$, which indicates that there were no significant differences in the predictive accuracy of the models. The interaction between measure type and model was significant, $F(5, 200) = 2.74$, $p < .001$. The interaction was largely driven by significant differences that indicated that the experimental models had better predictive ability than the null model for some of the measures, but there were no differences between the alternative models. For the

**Table 14. Post hoc tests for relative prejudice measures excluding D-scores (IAT Measures of Prejudice).**

| Comparison | | | | | |
| --- | --- | --- | --- | --- | --- |
| **Prejudice Measures** | **Mean Difference** | *SE* | *df* | *t* | *P* |
| All Measures – Actual | −0.01 | 0.01 | 40 | −1.13 | .265 |
| All Measures – Gut | −0.01 | 0.01 | 40 | −1.04 | .302 |
| All Measures – Positive | 0.02 | 0.02 | 40 | 1.15 | .258 |
| All Measures – Negative | 0.02 | 0.02 | 40 | 1.19 | .242 |
| All Measures – Preference | −0.03 | 0.01 | 40 | −3.67 | .001 |
| Actual – Gut | −0.002 | 0.01 | 40 | −0.15 | .880 |
| Actual – Positive | 0.03 | 0.02 | 40 | 1.69 | .099 |
| Actual – Negative | 0.03 | 0.01 | 40 | 1.71 | .095 |
| Actual – Preference | −0.02 | 0.01 | 40 | −1.81 | .078 |
| Gut – Positive | 0.03 | 0.02 | 40 | 1.53 | .135 |
| Gut – Negative | 0.03 | 0.02 | 40 | 1.56 | .126 |
| Gut – Preference | −0.02 | 0.01 | 40 | −1.80 | .080 |
| Positive – Negative | −0.0001 | 0.004 | 40 | −0.03 | .977 |
| Positive – Preference | −0.05 | 0.02 | 40 | −2.09 | .043 |
| Negative – Preference | −0.05 | 0.02 | 40 | −2.12 | .040 |

Table 14 contains the mean differences for the main effect of the type of relative prejudice measure.

**Table 15. Post hoc tests for model comparisons excluding D-scores (IAT measures of prejudice).**

| Comparison | | | | | |
| --- | --- | --- | --- | --- | --- |
| **Model** | **Mean Difference** | *SE* | *df* | *t* | *p* |
| Ideology – Status | −0.03 | 0.04 | 40 | −0.79 | .432 |
| Ideology – Choice | −0.02 | 0.04 | 40 | −0.50 | .618 |
| Ideology – Ideology + Status + Choice | −0.004 | 0.04 | 40 | −0.09 | .929 |
| Ideology – Null | −0.11 | 0.04 | 40 | −2.51 | .016 |
| Status – Choice | 0.01 | 0.04 | 40 | −0.29 | .773 |
| Status – Ideology + Status + Choice | 0.03 | 0.04 | 40 | 0.70 | .486 |
| Status – Null | −0.07 | 0.04 | 40 | −1.71 | .095 |
| Choice – Ideology + Status + Choice | 0.02 | 0.04 | 40 | 0.41 | .682 |
| Choice – Null | −0.09 | 0.04 | 40 | −2.00 | .052 |
| Ideology + Status + Choice – Null | −0.10 | 0.04 | 40 | −2.42 | .020 |

Table 15 contains the mean differences for the main effect of model type.

implicit measures, the effect of model type was significant, $F(4,15) = 4.43$, $p = .015$. The post-hoc tests revealed that the only significant differences between the models were between the null model and the alternative models, such that the null model performed worse than all other models. In the case of the pre-planned Study 2 analyses, and the analyses described here, we found that there were no significant differences in the models' predictive abilities (besides comparisons with the null model). Because we wrote in our Stage 1 manuscript that we would combine the measures, we do not interpret them strongly here.

## Study 2 discussion

In contrast to Study 1, there was no single model or measure (or combination of model and measure) that was a stronger predictor of the ideology-prejudice association than the other models and measures. However, despite the lack of

statistical significance, like Study 1, the MSEs for the ideology-only model and ideology, status, and choice model are descriptively the lowest. Additionally, it is important to note the differences in groups included in Study 1 versus Study 2. Study 1 included 18 individual target groups, whereas Study 2 included nine paired target groups for the explicit relative measures and six paired target groups for the IAT measure. Further, the three groups not included in the IAT measure are explicitly political (i.e., Democrat/Republican, Liberal/Conservative, and Religious/Atheist), which may have impacted the results. In our exploratory analyses when the D-score data were removed, the effect of measure type and the model × measure interaction was significant. This could suggest that including the most explicitly political groups in the analyses contributes to greater model accuracy. Additionally, it could suggest that the differences in model accuracy could be due to differences between explicit and implicit measures. The D-score measure may have added noise to the analyses in Study 2, as its removal seems to have contributed to greater model accuracy.

From a theoretical standpoint, some political groups are more explicitly political than others. For example, one might expect Liberals/Conservatives and Democrats/Republicans to be more politically relevant than Mother/Father or Foreign/Local people. Our exploratory analyses indicate that the models make the most different predictions for explicitly political groups as opposed to other groups. This is indicated by weaker predictive ability when explicitly political groups are excluded, compared to when they are included. We give suggestions for these potential limitations and suggestions for future directions in the general discussion.

Additionally, we ran exploratory analyses to examine how the separation of the explicit and implicit relative measures impacted the results. We did not state that we would do these additional analyses in our Stage 1 manuscript, but we felt it was an important set of analyses to consider given the moderate correlations between the explicit and implicit measures. We do not make strong claims about the results of these analyses, as they were not pre-registered or a part of the initial Stage 1 manuscript. Future researchers could test the differences between explicit and implicit measures in future model-building attempts. It is possible that explicit and implicit measures require different model-building strategies.

## General discussion

This work represented a stress-test of the models proposed by Brandt [1] using novel target groups, alternative explicit measures, and relative prejudice measures (including the IAT). In Study 1, we replicated Brandt's [1] findings in a different context and using different measures. This indicates that these models predict the ideology-prejudice association across different contexts. In the case of absolute measures, it seems to be the case that ideology alone as well as the combined perceptual dimensions of ideology, status, and choice are predictive of the ideology-prejudice association across measures and groups (particularly when explicitly political groups are included in model-testing). Because of this, we contend that future work can build on these existing models and add other perceptual dimensions or potentially even participant individual differences to increase their accuracy. Researchers can also use the predictive power of these models (specifically the ideology-only model) to make specific predictions of effect size for even broader numbers of targets, allowing for a stringent test of their hypotheses.

Here, we are placing more weight on the planned analyses described in our Stage 1 manuscript that contain the full range of available target groups. However, our exploratory analyses with explicitly political groups removed indicated that the accuracy of the ideology-only and ideology, status, and choice models may be due to their better predictive ability for these groups compared to the status-only and choice-only models. This is because the models do not make substantially different predictions for many of the other groups in the study. Other groups with strong ideological stereotypes (e.g., religious people, atheists; [1,38]) were not available in these data, but are groups where the models are likely to make divergent predictions.

In Study 2, we extended Brandt's [1] original work by developing new models using a different dataset. Although the ideology-only model was descriptively the most accurate, there were no significant differences in predictive accuracy between the different models and measure types. This may be explained by the smaller number of groups (pairs) included

in the analyses, as well as other factors described below. Alternatively, this potentially indicates that this strategy of model-building is not effective for relative measures, or that adjustments to this strategy would be needed to build effective models for relative measures.

### Limitations

These studies were not without their limitations. Our studies only included 18 groups in Study 1 and nine paired groups in Study 2. In contrast, Brandt [1] included 42 groups in his models. We included all possible groups in the available data, which included nine new groups not included in Brandt's [1] models. To best determine the generalizability of the models, more groups should be tested for both Brandt's [1] models and other possible competitor models.

Further, in Study 2, we were not able to use the D-score values for three target group pairs, Democrats/Republicans, Liberals/Conservatives, and Religious/Atheist. The IAT task for these groups (where participants compared themselves to the target groups) was different from the task for the other groups (associating groups with "good" and "bad" words), so we omitted those groups from the analyses. In addition, whereas Study 1 was based on preexisting models, Study 2 used a train/test split on the data to generate new models. It is possible the smaller number of participants in the test dataset generated enough noise to make it harder to detect effects. Future work can include relative measures to boost sample size as much as possible during the train/test process or potentially use a smaller training dataset and a larger test dataset during the train/test split.

### Future directions

The results of Study 1 indicate that Brandt's [1] models are generalizable to nine new groups, and to a large dataset outside of the one used to build them. To further demonstrate their generalizability, these models should be tested on more additional groups, beyond those we examine here. Additionally, given that we successfully replicated Brandt's [1] findings, we have evidence these models can be used by future researchers to generate hypotheses and estimate effect sizes regarding the ideology-prejudice association for various target groups.

Relatedly, the results of Study 2 indicate that it may be worth re-trying Brandt's [1] model-building strategy using more than nine group pairs to help develop a model of ideology and its association with relative prejudice. Although the MSE comparisons did not find differences in the model, the large differences in the exploratory correlations between the observed and predicted values for the ideology model compared to the status and choice model suggest that there may be meaningful differences in model accuracy yet to be discovered. This model-building strategy could be augmented with a larger sample of participants to precisely estimate associations. If this is not successful, it may indicate relative prejudice measures require a different model-building strategy to accurately predict the ideology-prejudice association for various target groups. Additionally, it may be worthwhile to consider including group characteristics beyond ideology, status, and choice in their predictive models, such as groups' perceived warmth and competence.

For both studies, it is useful to consider how the results might generalize outside of the United States. The idea behind the models is that ideological similarity and dissimilarity help explain the ideology-prejudice association across a wide range of target groups. When working in other contexts, it is necessary to use measures of ideology and groups that are relevant for that context. For example, Koch and colleagues' [38] methods for eliciting important target groups in different cultural contexts can be used to generate the relevant groups for the context. Similarly, a consultation with local experts and the relevant research literature can identify the most important ideological dimensions in the context. Although the liberal-conservative dimension may not be the most relevant outside of the US, alternatives like the left-right dimension, progressive-traditional dimension, or a secular-religious dimension appear in other country contexts (e.g., [39]). With context-relevant groups and ideology measures identified, scholars can follow the methods we've used here to develop and test the relevant models in alternative contexts. Although the precise ideological dimension and groups might differ,

we could expect that the ideological similarity and dissimilarity of groups would be a relevant factor in explaining the ideology-prejudice association.

## Conclusion

The current work indicates that the ideology-only model, as well as the ideology, status, and choice models are robust predictive tools for the relationship between ideology and absolute prejudice measures. Future researchers should consider using the ideology-only model to generate hypotheses and anticipate effect sizes for additional target groups and explore additional predictors and moderators. Although the results of Study 2 produced less conclusive estimates than those in Study 1, this work is still valuable because it provides clues that having groups with a wide range of perceived ideology is an important factor when generating predictive models of the ideology-prejudice relationship.

## Supporting information

**S1 Appendix. New Data Collection Survey Questions and Results.** This supplemental file includes feeling thermometers, status rating, ideology rating, and choice rating items for the new data collection described in the manuscript and the results.
(DOCX)

**S2 Appendix. Study 1 mixed ANOVA post hoc table.** This supplemental file includes the post hoc tests examining the Model x Measure interaction in the Mixed ANOVA.
(DOCX)

**S3 Appendix. Study 1 and study 2 analyses with demographic controls removed.** This supplemental file includes the output for the Study 1 and Study 2 analyses, with demographic controls removed from the multilevel models.
(DOCX)

**S4 Appendix. Study 1 mixed ANOVA exploratory results table.** This supplemental file includes the results for the exploratory mixed ANOVAs for the explicitly political groups and the other 14 groups.
(DOCX)

**S5 Appendix. Study 2 analyses with the explicit and implicit relative measures removed.** This supplemental file includes the results of the Study 2 analyses with the explicit and implicit relative measures separated.
(DOCX)

## Acknowledgments

We would like to thank Kathleen Schmidt and the Ideology 2.0 team for putting out the call for registered reports and for allowing us to use their data.

## Author contributions

**Conceptualization:** Jordan L. Thompson, Mark J. Brandt, Geoffrey A. Wetherell.

**Formal analysis:** Jordan L. Thompson, Mark J. Brandt, Geoffrey A. Wetherell.

**Funding acquisition:** Mark J. Brandt, Geoffrey A. Wetherell.

**Investigation:** Jordan L. Thompson, Mark J. Brandt, Geoffrey A. Wetherell.

**Methodology:** Jordan L. Thompson, Mark J. Brandt, Geoffrey A. Wetherell.

**Project administration:** Jordan L. Thompson.

**Resources:** Jordan L. Thompson, Abigail L. Cassario, Mark J. Brandt, Geoffrey A. Wetherell.

**Software:** Jordan L. Thompson, Sada Rice, Mark J. Brandt, Geoffrey A. Wetherell.

**Supervision:** Mark J. Brandt, Geoffrey A. Wetherell.

**Validation:** Jordan L. Thompson, Mark J. Brandt, Geoffrey A. Wetherell.

**Visualization:** Jordan L. Thompson, Mark J. Brandt, Geoffrey A. Wetherell.

**Writing – original draft:** Jordan L. Thompson.

**Writing – review & editing:** Jordan L. Thompson, Abigail L. Cassario, Shree Vallabha, Samantha A. Gnall, Sada Rice, Prachi Solanki, Alejandro Carrillo, Mark J. Brandt, Geoffrey A. Wetherell.

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
