## [Decision Letter · Decision Letter 0]

22 Jul 2025

PONE-D-25-07395Registered report: Stress testing predictive models of ideological prejudicePLOS ONE

Dear Dr. Thompson,

Thank you for submitting your manuscript to PLOS ONE. After careful consideration, we feel that it has merit but does not fully meet PLOS ONE’s publication criteria as it currently stands. Therefore, we invite you to submit a revised version of the manuscript that addresses the points raised during the review process.

 As you will find in the reviewer comments, both reviewers find merit in the research presented and both agree that the data may provide a useful contribution to the existing literature. However, both reviewers have concerns that we would like to see addressed in a final version of the paper. Because you have already published a Registered Report Protocol, you will likely need to address these comments to the best of your ability in the Discussion of your paper. Note that Reviewer 1 defers to me in a comment regarding the use of p values in addition to model fit. I recommend that you defer to the planned analyses as indicated in your published protocol (which includes the use of p values). Please feel free to reach out to me if you have any questions regarding the comments; I know it can get tricky when aspects of the paper are already published. 

We look forward to receiving your revised manuscript.

Kind regards,

Corey Cook

Academic Editor

PLOS ONE

Journal Requirements:

2. In your cover letter, please confirm that the research you have described in your manuscript, including participant recruitment, data collection, modification, or processing, has not started and will not start until after your paper has been accepted to the journal (assuming data need to be collected or participants recruited specifically for your study). In order to proceed with your submission, you must provide confirmation.

Reviewers' comments:

Reviewer's Responses to Questions

**Comments to the Author**

1. Does the manuscript adhere to the experimental procedures and analyses described in the Registered Report Protocol?

If the manuscript reports any deviations from the planned experimental procedures and analyses, those must be reasonable and adequately justified.

Reviewer #1: Yes

Reviewer #2: Yes

2. If the manuscript reports exploratory analyses or experimental procedures not outlined in the original Registered Report Protocol, are these reasonable, justified and methodologically sound?

A Registered Report may include valid exploratory analyses not previously outlined in the Registered Report Protocol, as long as they are described as such.

Reviewer #1: Yes

Reviewer #2: Yes

3. Are the conclusions supported by the data and do they address the research question presented in the Registered Report Protocol?

The manuscript must describe a technically sound piece of scientific research with data that supports the conclusions. The conclusions must be drawn appropriately based on the research question(s) outlined in the Registered Report Protocol and on the data presented.

Reviewer #1: Yes

Reviewer #2: Partly

4. Have the authors made all data underlying the findings in their manuscript fully available?

Reviewer #1: Yes

Reviewer #2: Yes

5. Is the manuscript presented in an intelligible fashion and written in standard English?

Reviewer #1: Yes

Reviewer #2: Yes

6. Review Comments to the Author

Please use the space provided to explain your answers to the questions above. (Please upload your review as an attachment if it exceeds 20,000 characters)

Reviewer #1: This was a paper extending some previous work building predictive models of prejudice toward various groups. I very much support iterative progression of these more formal modeling approaches, and I thought the adaptation to relative differences, another very common operationalization of prejudice in the field, was an important extension. Generally the authors find additional support for the ideology model over some other specifications. Thoughts and concerns follow:

Major concerns:

Because this paper is basically extending the Brandt 2017 model, some of these questions and critiques necessarily apply to that work as well. It's a little odd because that work is published and established, but still think it makes sense for the authors to engage with some of the questions in the present work.

-One thing I've always wondered with these models is about construct validity, or how much this "rating of ideology" is capturing the same or similar latent construct of prejudice. On their face, certainly distinct to me. But to the extent this variable is capturing "the extent to which i see these various groups on my team or not", it's less surprising to find such a strong link with how much I like those groups.

-A related concern is that these models depend on ratings on a bipolar liberal to conservative scale. Such a scale would just not work in the majority of countries in the world in which the political spectrum can't be reduced to one dimension. If our goal is purely non-explanation based prediction in the US, that's totally fine of course. But makes it more challenging to argue this is a causal model of the more universal force of prejudice, or it would have to be tweaked in some way. I guess this critique hinges on how the authors are interpreting this model, and I'd like to see that spelled out.

I've grouped these two together because they have perhaps been engaged with by the authors elsewhere, and I think could be handled here by some mention in the General Discussion.

-Regarding the modeling, I found the use of demographic controls conceptually problematic. If the goal is to predict prejudice from ideology, status, choice, and anything else, why should these be adjusted for the demographics of the participant? To adjust for gender, for example, is to say that men are just inherently more prejudiced toward xyz groups in a way that can't be explained or captured by other psychological variables, and I just don't believe that to be true. Plus it changes the very spare model (ideology) to be a lot more conceptually clunky, like who knows what other psychological baggage is being captured by "man" and "educated", but my guess is quite a bit. How much are results contingent on the inclusion of these demographic controls?

-My largest and perhaps controversial philosophical concern was the use of null hypothesis testing for model selection, and looking at p-values as evidence. I just didn't find it appropriate for the goals here. Models with the lowest average MSE simply fit the data best, regardless of whether they fit "significantly" better than other models with higher MSEs. Data science world would focus only on model fit, and this feels like a psych/data science mashup, employing both frameworks for evidence simultaneously. I'd defer to the editor, but I wouldn't be averse to cutting the p-value framework from the paper entirely, and relying only on model fit to determine the best models (and it would result in an identical conclusion from the line already being walked in the general discussion).

-What is an "ideologically relevant group" exactly? There needs to be a clear definition for this, rather than just sounds ideologically relevant to the authors and readers. I'm sympathetic in some respects, yes, those groups certainly seem ideologically relevant. But some of the groups not included in this bundle also seem ideologically relevant, and when I start squinting, many of these groups seem like they could also be ideologically relevant. A clear definition would sort this out, especially because the authors tenatively posit on page 25 that prediction success is better for these groups. Without some explanation, feel this risks being a "just so" story.

-On page 29, the authors mention they combined explicit and reaction time measures. What does this mean? How were they combined. Some folks consider implicit vs. explicit measures of prejudice quite distinct, so this feels odd.

Minor concerns:

-Intro of 2nd para- the prejudice literature has offered far more than just 3 predictors of prejudice. Maybe can revise this sentence to suggest Brandt identified three in his models. The entire field has prob put out at least 50.

-A line in the General Discussion mentions this is the "first time" predictive models were built for relative prejudice. While this is possibly true for the Brandt ideological prejudice models, it's not in general. Hehman & Neel psych review recently had some predictive models of prejudice with relative measures (and non-relative measures, a nice parallel to the current work and Brandt 2017). In general, I don't always find lines in papers noting "first" are so important, since that's somewhat obviated as the point of the paper, and could just be adjusted or trimmed.

-Lot of measures, would be great to have a correlation table between everything somewhere, apologies if I missed this.

I appreciate the opportunity to review this interesting work.

Reviewer #2: This paper presents a “stress test” of models for predicting the ideology-prejudice association, comparing models that incorporate the perceived ideology, status, and choice in group membership of various target groups. I have focused this review on the results, conclusions, and adherence to the initial registered report protocol, given that the methods and research question have been reviewed previously. Overall, the manuscript adheres to the procedures and analyses described in the protocol, and exploratory analyses are clearly marked. The authors were careful not to place too much weight on exploratory analyses relative to pre-planned analyses. However, my main concern with the manuscript is that the results of one of the exploratory analyses drastically alter the overall conclusions that I believe can be drawn from the manuscript. Despite this analysis not being pre-planned, I believe the changes to the results are sufficiently large that the authors should temper their conclusions in the paper to take them into account. I expand on this point below. If the conclusions and discussion can be revised to better incorporate the results of this analysis, I believe this manuscript will be well-suited for publication and provide an important contribution to the literature.

In my reading, the exploratory analysis reported on page 25 very significantly alters the results and warranted conclusions of the paper. The authors find that when they exclude targets groups that are defined by ideology (e.g., Democrats and Republicans), there are no differences among the predictiveness of the various models (including the null model). This seems highly significant to me because the claim that perceived ideology predicts prejudice is only theoretically meaningful or interesting insofar as it is about groups not explicitly defined by ideology. It seems trivial to say that the biggest predictor of how much Republicans like Democrats (and vice versa) is their perceived ideology – in fact these seem more like control groups than target groups to me. The results of the exploratory analysis indicate that the advantage of the ideology models over the status, choice, and null models is solely due to the inclusion of these groups that are explicitly defined by ideology. However, the authors’ discussion, conclusion, and abstract still make strong statements about the benefit of the ideology model(s) over the other models, which I believe are unwarranted given these results. I believe these conclusions and interpretations should be revised to reflect the fact that ideology provides no better prediction than the other models for the groups that are not defined only by ideology.

On a separate note, the logic of the analyses was not always intuitive, and I believe in some places could benefit from additional examples or explanation. At the top of page 20, for example, the authors discuss calculating a predicted ideology-prejudice association for the group “Black people”. Here, it would be very helpful to walk the reader through what each of the values represents – what does the final estimate of -0.49 in this context mean? Further, the authors at times refer to predicting prejudice and at other times to the ideology-prejudice association, and I think this distinction could be clarified and highlighted more. As I understand, the main purpose of the study is to predict the association between ideology and prejudice; however, on page 20, for example, the authors seem to be working with actual prejudice values rather than associations (“we subtracted the actual ratings of prejudice across groups from the estimates from the equations, squared these values, and then found the average”). Additional explanation would be helpful here – why are actual prejudice values rather than associations being predicted here? Finally, at times the distinction between the participant’s ideology and the perceived ideology of the group became confusing, and I would encourage the authors to explicitly state which of the two they are referring to when they use the word “ideology” in order to reduce cognitive load for the reader.

Below are a few additional minor questions:

1. Could the authors expand on the choice to use mean squared error as a metric to compare models, rather than a metric that takes into account the number of parameters included in the model?

2. Were all the demographic control variables in the models used to get the observed ideology-prejudice associations the same as the ones used by Brandt (2017) to get the predicted ones?

7. PLOS authors have the option to publish the peer review history of their article (what does this mean? ). If published, this will include your full peer review and any attached files.

**Do you want your identity to be public for this peer review?** For information about this choice, including consent withdrawal, please see our Privacy Policy .

Reviewer #1: No

Reviewer #2: No

---

## [Author Response · Author response to Decision Letter 1]

22 Aug 2025

We thank the editor for his work on our manuscript. We have addressed the reviewers’ concerns primarily in the discussion and conclusion sections (but we have added a few clarifications in the introduction, see page 3). Additionally, we have retained our use of p-values to maintain consistency with our Stage 1 manuscript.

We wanted to mention that in double-checking our code, we noticed that the values for status and choice for Black people were switched by mistake. We have re-run our analyses and found the same pattern of results. The values in the manuscript have been updated to reflect this, and they have changed very slightly. We apologize for this error, and we have double-checked the other values, and they are correct.

Please note, as requested in the decision email, we have added two new references. To strengthen our discussion, we have added the references: Koch et al., 2016 and Yustisia et al., 2025.

General Response From the Authors

We thank the editor and reviewers for their close attention to our manuscript. We respond to each point below. Reviewer comments are in plain text, and our responses are in italics. We also include the relevant sections of the paper in quotes for easy reading. These are in quotes, in italics, and with bullet points.

Reviewer 1

This was a paper extending some previous work building predictive models of prejudice toward various groups. I very much support iterative progression of these more formal modeling approaches, and I thought the adaptation to relative differences, another very common operationalization of prejudice in the field, was an important extension. Generally the authors find additional support for the ideology model over some other specifications. Thoughts and concerns follow:

Major concerns:

Because this paper is basically extending the Brandt 2017 model, some of these questions and critiques necessarily apply to that work as well. It's a little odd because that work is published and established, but still think it makes sense for the authors to engage with some of the questions in the present work.

-One thing I've always wondered with these models is about construct validity, or how much this "rating of ideology" is capturing the same or similar latent construct of prejudice. On their face, certainly distinct to me. But to the extent this variable is capturing "the extent to which i see these various groups on my team or not", it's less surprising to find such a strong link with how much I like those groups.

We thank the reviewer for bringing up this point. We can see where the reviewer is coming from. However, we do not think that these ideology ratings are capturing the latent concept of prejudice. Our models include two instances of “ideology” – perceived group ideology (and/or status and choice for the other models) and participant ideology. The perception of groups’ ideological leanings is largely consistent and stable (our values were similar to Brandt, 2017 and Brandt & Crawford, 2016). The literature also shows that people have a consensus about a groups’ perceived ideology and agency/competence (similar to status; Koch et al., 2020). In contrast, perceived warmth (closer to our prejudice measure) is not agreed upon (Koch et al., 2020). If perceived ideology and prejudice were tapping into the same latent construct, they should be similarly consensually perceived. As the reviewer alludes to, our measures are highly face-valid, and are commonly used across literatures (e.g., Crawford et al., 2017, Jost, 2006; Jost, 2017). The constructs of one’s ideology and one’s prejudice toward certain groups are related, but they do not completely overlap. The ideological values used in our equations are related to the stable perceptions of groups’ ideologies as opposed to individual participants’ ideological positions.

We would also like to point out that the measures of status and choice are much less likely to have the type of shortcoming the reviewer brings up. Our study is also relevant to these perceptions as it shows that these group perceptions do not help predict the ideology-prejudice association. This is notable as theories in social and political psychology suggest that the ideology-prejudice association might be better explained by these variables. We show that not only is it not better explained, it is not explained by status and choice.

-A related concern is that these models depend on ratings on a bipolar liberal to conservative scale. Such a scale would just not work in the majority of countries in the world in which the political spectrum can't be reduced to one dimension. If our goal is purely non-explanation based prediction in the US, that's totally fine of course. But makes it more challenging to argue this is a causal model of the more universal force of prejudice, or it would have to be tweaked in some way. I guess this critique hinges on how the authors are interpreting this model, and I'd like to see that spelled out.

I've grouped these two together because they have perhaps been engaged with by the authors elsewhere, and I think could be handled here by some mention in the General Discussion.

This is a good point. We have added a discussion of the generalizability of our results in the Future Directions section (see pages 48-49).

• “For both studies, it is useful to consider how the results might generalize outside of the United States. The idea behind the models is that ideological similarity and dissimilarity help explain the ideology-prejudice association across a wide range of target groups. When working in other contexts, it is necessary to use measures of ideology and groups that are relevant for that context. For example, Koch and colleagues’ [39] methods for eliciting important target groups in different cultural contexts can be used to generate the relevant groups for the context. Similarly, a consultation with local experts and the relevant research literature can identify the most important ideological dimensions in the context. Although the liberal-conservative dimension may not be the most relevant outside of the US, alternatives like the left-right dimension, progressive-traditional dimension, or a secular-religious dimension appear in other country contexts (e.g., [40]). With context-relevant groups and ideology measures identified, scholars can follow the methods we’ve used here to develop and test the relevant models in alternative contexts. Although the precise ideological dimension and groups might differ, we could expect that the ideological similarity and dissimilarity of groups would be a relevant factor in explaining the ideology-prejudice association.”

-Regarding the modeling, I found the use of demographic controls conceptually problematic. If the goal is to predict prejudice from ideology, status, choice, and anything else, why should these be adjusted for the demographics of the participant? To adjust for gender, for example, is to say that men are just inherently more prejudiced toward xyz groups in a way that can't be explained or captured by other psychological variables, and I just don't believe that to be true. Plus it changes the very spare model (ideology) to be a lot more conceptually clunky, like who knows what other psychological baggage is being captured by "man" and "educated", but my guess is quite a bit. How much are results contingent on the inclusion of these demographic controls?

Thank you for this question. To address this, we ran the analyses in Study 1 and Study 2 without the demographic controls (see S3 Appendix and pages 27 and 40 of the manuscript). Regarding our inclusion of demographic controls, we included them for two reasons. First, to conceptually capture the relationship between ideology and prejudice (as opposed to predicting prejudice) in isolation over and above prejudices potentially related to participant traits (e.g., Black participants liking Black people and vice-versa for White participants). Second, Brandt (2017) included demographic controls, and our intention in Study 1 was to replicate his model testing process as closely as possible. We found the same results patterns with and without the demographic controls.

• Page 27: “For the curious reader, we also ran the Study 1 analyses without the demographic control variables and found the same pattern of results (see S3 Appendix). As our Stage 1 report was accepted with the inclusion of controls in our analysis plan, we do not interpret those results further in-text.”

• Page 40: “As we did in Study 1, we also ran the Study 2 analyses without the demographic control variables and found the same pattern of results (see S3 Appendix). We interpreted the analyses with the controls here, as this is consistent with our Stage 1 manuscript, and our intention in Study 2 is to run analyses that parallel Study 1.”

-My largest and perhaps controversial philosophical concern was the use of null hypothesis testing for model selection, and looking at p-values as evidence. I just didn't find it appropriate for the goals here. Models with the lowest average MSE simply fit the data best, regardless of whether they fit "significantly" better than other models with higher MSEs. Data science world would focus only on model fit, and this feels like a psych/data science mashup, employing both frameworks for evidence simultaneously. I'd defer to the editor, but I wouldn't be averse to cutting the p-value framework from the paper entirely, and relying only on model fit to determine the best models (and it would result in an identical conclusion from the line already being walked in the general discussion).

We thank the reviewer for raising this concern. In the Stage 1 manuscript, we discussed using p-values as a means of determining the model with the best fit. The editor has suggested that we use p-values to remain consistent with the Stage 1 plan, so we have retained our use of p-values, although we do acknowledge the controversies surrounding using them (e.g., Wasserstein & Lazar, 2016). To augment our MSE/p-value interpretations, we ran correlations between the predicted and observed values for each model in both studies (see Table 1, page 7, and Table 11, page 36 for their additions to this manuscript). In both studies, the ideology-only (r = .97, .92) and ideology, status, and choice models (r = .96, .92) had higher correlation coefficients than the status-only (r = .45, .43) and choice-only (r = .17, .62) models.

-What is an "ideologically relevant group" exactly? There needs to be a clear definition for this, rather than just sounds ideologically relevant to the authors and readers. I'm sympathetic in some respects, yes, those groups certainly seem ideologically relevant. But some of the groups not included in this bundle also seem ideologically relevant, and when I start squinting, many of these groups seem like they could also be ideologically relevant. A clear definition would sort this out, especially because the authors tenatively posit on page 25 that prediction success is better for these groups. Without some explanation, feel this risks being a "just so" story.

We appreciate the reviewer’s request for clarification. We have added a clearer explanation for which groups can be considered “ideologically-relevant” on page 28. We also changed the term to “explicitly political groups.” The groups we flag in our discussion of the results are explicitly based on ideological beliefs and political identities. We now note this.

• “Although the patterns in our planned analyses replicated Brandt [1], we explored whether our models were more predictive of the ideology-prejudice association for explicitly political groups. This seemed relevant to us given the results of the relative measures models reported below (i.e., results were less strong in the subset of measures that did not include explicitly political groups). While many of the included groups can be considered political to some extent, some groups are explicitly political and tied to liberal-conservative politics in the United States (i.e., liberals, conservatives, Democrats, Republicans).”

-On page 29, the authors mention they combined explicit and reaction time measures. What does this mean? How were they combined. Some folks consider implicit vs. explicit measures of prejudice quite distinct, so this feels odd.

This is an important clarification. We have updated our description on page 33. To further clarify, we used all the implicit and explicit prejudice measures available in the data set. In order to run our multilevel models, we turned the data long (i.e., such that each participant has multiple rows organized by prejudice measure). When we turned the data long, we created our prejudice outcome variable, which represents both the explicit and implicit prejudice measures(see lines 268-282 in the Study 2 code). In the context of our multilevel models, this means the measures were not combined per se but were nested within participants in separate rows. Given the moderate correlations (i.e., ranging from .35 - .45) between the explicit and implicit measures, this seems justified as these correlations suggest the relationship between these measures is non-trivial.

At the same time, we do see the reviewer’s concern. To address concerns about combining the measures, we ran additional analyses with the explicit and implicit measures separated. The resulting predictive equations were different (see the S5 Appendix), but neither found any difference between the ideology, status or choice models (there is some limited evidence that all models did better than the null model). However, given that we wrote in our Stage 1 manuscript that we would combine the measures, we have decided to keep this in the manuscript. We also note our analyses with the measures separated on pages 43-44 and recommend that future researchers could test if separating the measures helps better predict the size and direction of the ideology-prejudice association (page 45).

• Page 33: “In our analyses, we used both the explicit relative measures and the reaction time outcomes because they are measures of comparative prejudice (they were also moderately correlated, see Table 3). Specifically, these measures both involve making a choice to indicate preference for one group over another. In the context of our multilevel models, this means each measure was nested within participants in separate rows.”

• Pages 43-44: “Per a reviewer’s suggestion, we also ran the Study 2 analyses with the explicit and implicit relative measures separated. Although the explicit and implicit measures were moderately correlated (ranging from .35 - .45, see Table 3), these correlations indicate potential differences between the measures. For full results, see the S5 Appendix. Most importantly, the effect of model type was not significant, F(4, 40) = 1.43, p = .243, which indicates that there were no significant differences in the predictive accuracy of the models. The interaction between measure type and model was significant, F(5, 200) = 2.74, p < .001. The interaction was largely driven by significant differences that indicated that the experimental models had better predictive ability than the null model for some of the measures, but there were no differences between the alternative models. For the implicit measures, the effect of model type was significant, F(4,15) = 4.43, p = .015. The post-hoc tests revealed that the only significant differences between the models were between the null model and the alternative models, such that the null model performed worse than all other models. In the case of the pre-planned Study 2 analyses, and the analyses described here, we found that there were no significant differences in the models’ predictive abilities (besides comparisons with the null model). Because we wrote in our Stage 1 manuscript that we would combine the measures, we do not interpret them strongly here.”

• Page 45: “Additionally, we ran exploratory analyses to examine how the separation of the explicit and implicit relative measures impacted the results. We did not state that we would do these additional analyses in our Stage 1 manuscript, but we felt it was an important set of analyses to consider given the moderate cor

---

## [Editor Report · Decision Letter 1]

15 Sep 2025

PONE-D-25-07395R1Registered report: Stress testing predictive models of ideological prejudicePLOS ONE

Dear Dr. Thompson,

Thank you for your resubmission to PLOS ONE. I think you've done an excellent job addressing the reviewers' concerns, and I think your final manuscript will make a nice contribution to the journal and to the literature. Please consider your manuscript conditionally accepted (although I had to select 'minor revision' to solicit final edits). Once you submit a clean, unmasked version, I will formally accept the manuscript. (One pedantic side note: in a few places you reference "this data." Please correct to "these data" as "data" are plural; I shouldn't care, but I hound my Methods students about this point, so I felt compelled to note it here.)

You can ignore some of the boilerplate instructions below (e.g., a rebuttal letter), which pertain specifically to resubmissions. But please let me know if you have any questions or issues.

We look forward to receiving your revised manuscript.

Kind regards,

Corey Cook

Academic Editor

PLOS ONE
---

## [Author Response · Author response to Decision Letter 2]

16 Sep 2025

Thank you for your positive feedback! We have created a clean copy of the manuscript with instances of “blinded for review” removed and instances of “this data” changed to “these data.”

---

## [Editor Report · Decision Letter 2]

23 Sep 2025

Registered report: Stress testing predictive models of ideological prejudice

PONE-D-25-07395R2

Dear Dr. Thompson,

We’re pleased to inform you that your manuscript has been judged scientifically suitable for publication and will be formally accepted for publication once it meets all outstanding technical requirements.

Kind regards,

Corey Cook

Academic Editor

PLOS ONE

---

## [Editor Report · Acceptance letter]

PONE-D-25-07395R2

PLOS ONE

Dear Dr. Thompson,

I'm pleased to inform you that your manuscript has been deemed suitable for publication in PLOS ONE. Congratulations! Your manuscript is now being handed over to our production team.

Kind regards,

on behalf of

Dr. Corey Cook

Academic Editor

PLOS ONE